# Fungicidal Protection as Part of the Integrated Cultivation of Sugar Beet: An Assessment of the Influence on Root Yield in a Long-Term Study

Iwona Jaskulska [1], Jarosław Kamieniarz [2], Dariusz Jaskulski [1,*], Maja Radziemska [3,4] and Martin Brtnický [4]

1. Department of Agronomy, Faculty of Agriculture and Biotechnology, Bydgoszcz University of Science and Technology, 7 Prof. S. Kaliskiego St., 85-796 Bydgoszcz, Poland; jaskulska@pbs.edu.pl
2. Skalista 16, 62-080 Sierosław, Poland; jaroslawkamieniarz@gmail.com
3. Institute of Environmental Engineering, Warsaw University of Life Sciences, Nowoursynowska 159, 02-776 Warsaw, Poland; maja_radziemska@sggw.edu.pl
4. Department of Agrochemistry, Soil Science, Microbiology and Plant Nutrition, Faculty of AgriSciences, Mendel University in Brno, Zemedelska 1, 61300 Brno, Czech Republic; martin.brtnicky@seznam.cz
* Correspondence: darekjas@pbs.edu.pl

**Abstract:** Despite the major role of non-chemical treatments in integrated plant protection, fungicides often need to be applied as a crop protection treatment in sugar beet farming. They should be used based on a good understanding of the requirements and effectiveness of the active ingredients. In 11-year field experiments, the effect that one and three foliar applications of fungicides containing various active ingredients (triazoles, benzimidazoles, strobilurines) had on sugar beet root yields was assessed, depending on various thermal and rainfall conditions. It was found that in eight of the 11 years, foliar application of fungicides increased yields compared to unprotected plants, and three foliar treatments during the growing season were more effective than a single application. The negative correlation of the root yield of fungicidally protected plants with total June rainfall was weaker than the same relationship for unprotected plants. At the same time, the positive correlation between the yield of fungicidally protected sugar beets and average June air temperature was stronger than the same relationship for unprotected plants. The research results indicate the need to conduct long-term field experiments and to continuously improve integrated production principles for sugar beet, especially regarding the rational use of pesticides.

**Keywords:** *Beta vulgaris* L.; fungicides; plant protection; precipitation conditions; thermal conditions; multi-year experiment

## 1. Introduction

Sugar beet (*Beta vulgaris* L. *subsp. vulgaris var. altissima* Döll.) is one of the two most important sugar plants and has great economic and environmental potential, and Poland is one of its largest producers in Europe [1–3]. Root yield is shaped by genotype, environment, and agricultural technology alike. In the face of the numerous pathogens that threaten this plant, including Cercospora leaf spot (*Cercospora beticola*), Ramularia leaf spot (*Ramularia beticola*), and beet powdery mildew (*Erysiphe betae*), plant protection is an important agricultural technology [4,5]. Integrated pest management is currently the preferred method for maintaining the phytosanitary condition of agrocenosis, and although it ascribes a dominant role to breeding and agrotechnical treatments [6–9], it still holds an important place for chemical crop protection products, including fungicides [10–12]. However, the use of fungicides should be preceded by an assessment of the risk of plant infection or the presence and harmfulness of the pathogen [13,14].

In the last two decades, numerous active ingredients of foliar fungicides belonging to several chemical groups (mainly triazoles and strobilurins) have been tested in the protection of sugar beet. These include mencozeb, thiophanate methyl, triphenyltin hydroxide, azoxystrobin, tetraconazole, a mixture of propiconazole and trifloxystrobin, pyraclostrobin, fenbuconazole, and others [15–17]. However, their effectiveness and efficiency depend on many biological, environmental, and agrotechnical factors. Weather patterns are important, as they affect air humidity and temperature, as well as the duration of the period for which leaf blades are wet. Other important determinants of plant protection effectiveness are the number and method of fungicide treatments during the growing season and the selection of active ingredients in successive treatments [18–20]. Diversifying active ingredients can reduce the risk of pathogens developing fungicide resistance [21].

The application of foliar fungicides in integrated sugar beet cultivation beneficially affects not only the health of leaves and the leaf area index but also the yield and technological quality of roots [22–24]. Therefore, and also in view of the preference for choosing agricultural technologies that accord with the principles of integrated cultivation, long-term research was carried out on the impact that various fungicidal protection methods had on sugar beet root yields. It was assumed that only in the long term (over 10 years) was there a high probability of diverse environmental conditions (i.e., meteorological conditions) occurring that would allow for the assessment of the effectiveness of various active ingredients, the legitimacy of using several treatments during the growing season; and in particular the need for and effectiveness of different variants of protection fungicide depending on weather patterns.

The aim of the study was to determine the effect that one and several treatments involving the foliar application of fungicides containing various active ingredients had on sugar beet root yields depending on various thermal and precipitation conditions in 11-year field experiments.

## 2. Materials and Methods

A single-factor, long-term (11-year) field experiment was conducted in the years 2006–2016 at the Experimental Unit of Nordzucker Polska S.A. in the Chełmża plantation area (18°37′ E, 53°11′ N, 91 m a.s.l.) in Kuyavia-Pomerania Voivodeship. The studied treatments in the field experiment were active ingredients of foliar fungicides from various groups used to protect sugar beet and number of sprayings during the growing season: control (without fungicide), three applications, each containing a different active ingredient (triazoles, benzimidazoles, strobilurins), one application (tebuconazole), one application (epoxyconazole), one application (epoxyconazole + thiophanate-methyl), one application (strobilurin). Each experimental treatment was performed in randomized blocks in four repetitions (four plots, each 2.5 m × 10 m).

The experiments were conducted on Cambisol soils [25]. The content of organic carbon in the soil varied between years and between study sites, ranging from 7.4 to 11.7 g $C_{org}$ kg soil$^{-1}$, while assimilable forms of macronutrients (mg P, K, Mg kg soil$^{-1}$) varied in the respective ranges 69.0–138.1; 145.8–266.7; 37.0–100, and pHKCL was 5.8–6.9. The field experiments were performed under a continental climate characterized as Dfb [26]. In this region, the average annual precipitation is around 500 mm, and the average air temperature is around 8 °C. The minimum and maximum average monthly air temperatures are −5.2 °C and 20.1 °C, respectively. Meteorological data for the study years and long-term data come from the meteorological station in Falęcin, which lies within the same general area as the field experiments. The characteristics of thermal and precipitation conditions during the sugar beet growing season in the study years are below—chapter Results.

The forecrop for sugar beet was winter wheat each study year, and beet was cultivated, in rotation, once every 5–6 years. The soil was conventionally tilled: plowing with a combination of post-harvest tillage and pre-winter deep plowing. Phosphorus-potassium fertilizer was applied according to the levels of assimilable forms of nutrients in the soil.

Nitrogen fertilization of 120 kg N ha$^{-1}$ was divided into two doses, a pre-sowing dose of 60 N ha$^{-1}$ and a top-dressing dose of 60 kg N ha$^{-1}$ applied up to the BBCH 39 phase. The sowing date ranged, depending on the research year, between 3 April at the earliest (in 2009) and 24 April at the latest (in 2008 and 2013). According to progress in breeding, different cultivars were sown during the research period. These were, chronologically: "Kujawska"—four years, "Jagoda"—one year, "Pewniak"—one year, "Schubert"—one year, "Socrates"—one year, "Sinan"—two years, "Janpol"—one year.

The occurrence of dicotyledonous and monocotyledonous weeds was reduced by careful cultivation of the soil and then chemically by herbicides in split doses. The active ingredients used were: chloridazone, lenacil, metamitron, desmediphan, ethofumesate, phenmediphan, triflusulfuron-methyl, haloxyfop-P. Pests were controlled as interventions once harmfulness thresholds had been exceeded and using active ingredients registered for use in a given period, e.g., deltamethrin, dimethoate. Foliar fungicides were applied according to the experimental treatments. A single fungicide treatment was applied after the first symptoms of leaf spot (*Cercospora beticola*) appeared. Depending on the research year, this was between 27 July and 1 September. For the three fungicide applications, the first was performed around mid-July, the second 10–14 days later, and the third 3–4 weeks after the fungicide was applied on the plots subjected to only one application. More detailed information on the use of fungicides (e.g., products, dosages) is presented in an earlier article. The cited study also contains detailed information on the habitat and agrotechnical conditions in which the experiments were carried out and the main results, such as pathogen infestation and leaf damage, LAI (leaf area index), and leaf yield. Sugar beet leaves were infected by *Cercospora beticola*, *Ramularia beticola*, *Erysiphe betae*, and, to a lesser extent, *Uromyces beticola*. Leaf damage by pathogens was least after three applications of fungicides. The degree of damage to leaves of unprotected plants was significantly positively correlated with sums of precipitation in April–July and June–August [27]. Roots were harvested and evaluated between 12 October (2015) and 6 November (2006 and 2008). Beetroots were collected from the entire surface of each plot and weighed after the removal of impurities. The final root yield was expressed in t ha$^{-1}$. The yield was analyzed in relation to the applied fungicides—the experimental treatments. The relationship between yield and precipitation, and air temperature in individual months of sugar beet vegetation was also assessed. The assessment of root quality characteristics (biological content of sucrose, content of molasses-forming compounds, and sugar yield) and the determination of the dependence of these characteristics on the method of plant protection and its interaction with environmental conditions will be addressed in the second part of the scientific study.

Root yield, its variability over the years, and its correlation with hydrothermal conditions (i.e., precipitation and air temperature during the beet growing season) were statistically processed. One-way analysis of variance was performed. The significance of the influence of the experimental factor (the F statistic) and the significance of differences in means between treatments was assessed using Tukey's post-hoc test at $p = 0.05$. The variability in root yield between study years for the different studied treatments was assessed using the coefficient of variation (CV), and trends of changes in yield were estimated by linear regression. Meanwhile, multiple regression with the elimination of non-significant terms was used to assess the dependence of sugar beet root yield on thermal and precipitation conditions. The strength of root yield's relationship with average air temperature and the sum of precipitation was determined using Pearson's simple correlation. In addition to the statistical significance of the correlation coefficient $r$, the following classification of $|r|$ was adopted to assess its strength: 0.0–0.3 weak correlation; 0.3–0.5 moderate correlation; 0.5–0.7 strong correlation; 0.7–1.0 very strong correlation. Microsoft Excel 2016 [28] and Statistica 12 [29] were used.

## 3. Results

Average monthly air temperatures during the beet growing season in the study years ranged from 6.6 °C in October to 22.1 °C in July. The air temperature was most variable

at the sugar beet growing season's beginning (April) and end (September and October) (Table 1). Average precipitation in the months of beet vegetation in the study period was close to long-term averages (Table 2). The most variable in terms of monthly precipitation were April, September, and October (CV 61.6–100.4%), whereas precipitation was slightly more uniform from May to August (CV 44.3–56.8%).

**Table 1.** Characteristics of thermal conditions during the sugar beet growing season in the years 2006–2016 and in the long-term period (average monthly air temperature—°C as the "t" variable in the regression analysis).

| Characteristic | Month | | | | | | |
| --- | --- | --- | --- | --- | --- | --- | --- |
| | April | May | June | July | August | September | October |
| | Variable in the Regression Analysis | | | | | | |
| | tIV | tV | tVI | tVII | tVIII | tIX | tX |
| Minimum | 7.7 | 12.7 | 15.0 | 18.3 | 17.3 | 8.8 | 6.6 |
| Maximum | 11.2 | 15.6 | 18.9 | 22.1 | 21.7 | 16.1 | 11.0 |
| Mean (2002–2016) | 9.2 | 14.0 | 16.9 | 19.6 | 18.6 | 13.7 | 8.6 |
| Standard deviation | 1.3 | 1.0 | 1.2 | 1.5 | 1.2 | 2.1 | 1.3 |
| Coefficient of variation (%) | 14.6 | 7.3 | 7.3 | 7.4 | 6.4 | 15.3 | 15.5 |
| Long-term mean | 8.9 | 14.1 | 16.7 | 19.1 | 18.6 | 13.6 | 8.7 |

**Table 2.** Characteristics of precipitation conditions during the sugar beet growing season in the years 2006–2016 and in the long-term period (monthly sum of precipitation [mm] as the "o" variable in the regression analysis).

| Characteristic | Month | | | | | | |
| --- | --- | --- | --- | --- | --- | --- | --- |
| | April | May | June | July | August | September | October |
| | Variable in the Regression Analysis | | | | | | |
| | oIV | oV | oVI | oVII | oVIII | oIX | oX |
| Minimum | 1.2 | 16.9 | 23.4 | 34.8 | 7.5 | 0.1 | 5.3 |
| Maximum | 57.0 | 159.7 | 140.6 | 198.1 | 161.7 | 78.6 | 139.0 |
| Mean (2006–2016) | 27.6 | 66.7 | 62.3 | 114.7 | 87.6 | 35.3 | 40.0 |
| Standard deviation | 17.0 | 37.0 | 35.4 | 50.8 | 48.4 | 24.4 | 40.2 |
| Coefficient of variation (%) | 61.6 | 55.4 | 56.8 | 44.3 | 55.2 | 69.1 | 100.4 |
| Long-term mean | 29.0 | 67.0 | 59.3 | 121.8 | 74.9 | 40.9 | 41.5 |

In eight out of eleven study years, fungicidal protection had a significant effect on sugar beet root yield, whereas no such effect was found in only three years (Table 3). In one year, every active substance and the application of three treatments during the growing season resulted in an increase in root yield, and, in 2010 only, the three-time application of fungicides was more efficient and had a better effect on yield than a single treatment. Compared to the unprotected control plot, the root yield increased significantly under the influence of all active substances in two years, i.e., in 2007 and 2012.

On average, throughout the study period, the effect of all active ingredients used in one treatment on root yield was statistically equally beneficial and significant (Figure 1). However, an even higher yield was obtained after applying three fungicide treatments.

Fungicidal protection of sugar beet reduced root yield variability over the 11-year research period. The difference in the CV coefficient compared to the unprotected plot ranged from 0.4 to 2.3 percentage points and was seen for all active ingredients except strobilurin (Figure 2).

**Table 3.** Root yield (t·ha$^{-1}$) in the study years, by fungicidal protection method.

| Year | Method of Protection | | | | | |
|---|---|---|---|---|---|---|
| | Control | Three Treatments | Tebuconazole | Epoxiconazole | Epoxiconazole + Thiophanate-Methyl | Strobilurin |
| 2006 | 70.2 | 69.9 | 71.4 | 68.6 | 69.7 | 69.9 |
| 2007 | 69.4 c * | 84.1 a | 77.5 b | 79.6 ab | 79.2 ab | 76.1 b |
| 2008 | 71.9 b | 83.1 a | 82.8 a | 79.2 a | 76.9 ab | 82.4 a |
| 2009 | 55.3 bc | 62.8 a | 61.1 a | 60.1 ab | 58.5 ab | 52.4 bc |
| 2010 | 75.5 b | 89.7 a | 80.0 b | 79.0 b | 80.8 b | 77.9 b |
| 2011 | 67.4 | 72.6 | 71.9 | 71.7 | 68.8 | 69.8 |
| 2012 | 60.5 c | 72.8 ab | 67.4 b | 69.1 ab | 71.2 ab | 73.7 a |
| 2013 | 84.9 b | 92.5 a | 84.4 b | 87.9 ab | 92.0 a | 93.6 a |
| 2014 | 84.8 ab | 89.1 a | 85.7 ab | 81.3 b | 84.2 ab | 80.1 b |
| 2015 | 67.8 | 72.5 | 67.6 | 70.1 | 72.9 | 72.2 |
| 2016 | 82.8 b | 92.0 a | 83.6 b | 90.3 a | 87.3 ab | 81.9 b |

* letters in cells indicate statistically significant differences (Tukey's test at $p = 0.05$).

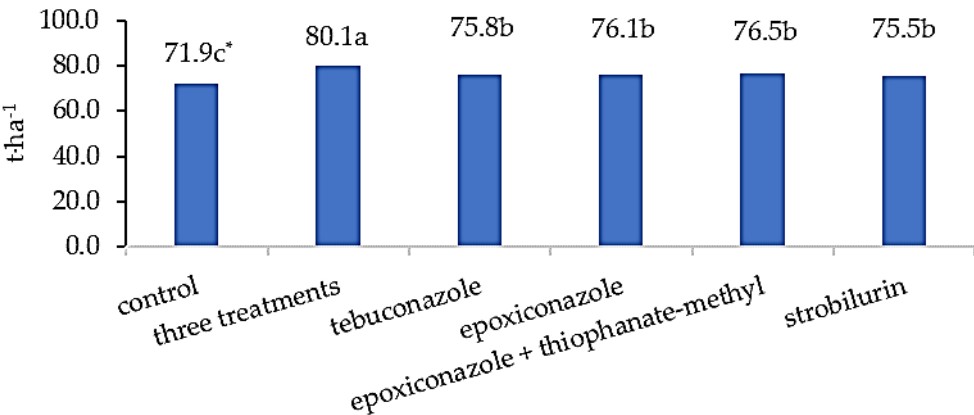

**Figure 1.** Average root yield in 2006–2016, by fungicidal protection method (* letters indicate statistically significant differences, Tukey's test at $p = 0.05$).

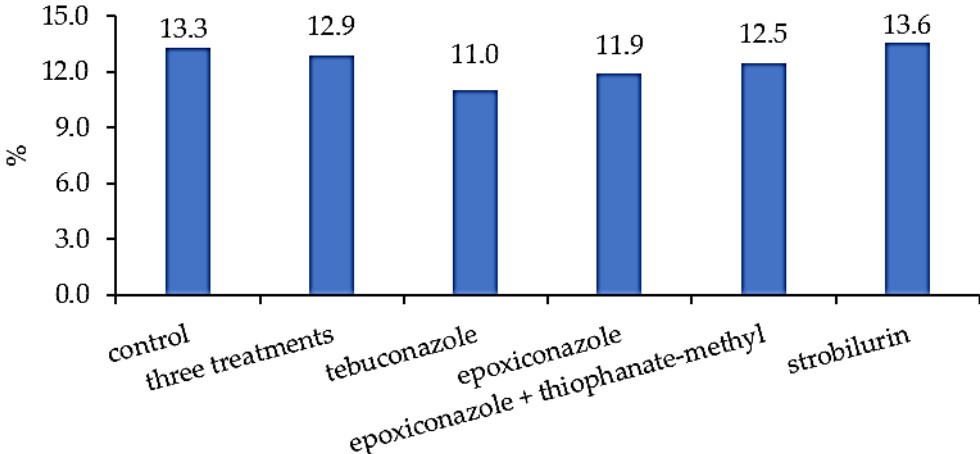

**Figure 2.** Variation coefficient of root yield in 2006–2016, by fungicidal protection method.

The upward trend in yields of protected plants in successive years was lower than that of unprotected ones (Table 4). On the control plot, according to the simple regression equation, the root yield increased by 1.27 t·ha$^{-1}$ each year. On protected plots, this increase ranged from 0.58 t·ha$^{-1}$ for tebuconazole to 1.13 t·ha$^{-1}$ for three fungicide treatments. Only for the mixture of epoxiconazole + thiophanate methyl, was the increase in yield similar to that of unprotected plants.

**Table 4.** Trend of changes in root yield in study years by fungicidal protection method.

| Protection Method | Regression Equation | Correlation Coefficient | Determination Coefficient |
|---|---|---|---|
| Control | $^\$$ y1 = 1.27 $^\&$ x + 64.27 | 0.438 | 0.192 |
| Three treatments | y1 = 1.13x + 73.33 | 0.363 | 0.132 |
| Tebuconazole | y1 = 1.13x + 72.27 | 0.232 | 0.054 |
| Epoxiconazole | y1 = 1.13x + 69.35 | 0.411 | 0.169 |
| Epoxiconazole + Thiophanate-methyl | y1 = 1.13x + 68.76 | 0.450 | 0.202 |
| Strobilurin | y1 = 1.13x + 69.13 | 0.341 | 0.116 |

$^\$$ y1—root yield (t·ha$^{-1}$); $^\&$ x—year of study.

The root yield of unprotected plants correlated most positively (r > 0.500) with the average air temperature in June and most negatively with the September temperature (r < −0.500). In both cases, but especially for the June temperature, fungicidal protection generally increased the strength of this relationship (Table 5). After the application of epoxiconazole and tebuconazole, the correlation of yield with average June and September temperatures, respectively, was statistically significant.

**Table 5.** Pearson's simple correlation coefficients between root yields and average monthly air temperature during the sugar beet growing season, by fungicidal protection method.

| Protection Method | Month | | | | | | |
|---|---|---|---|---|---|---|---|
| | April | May | June | July | August | September | October |
| Control | −0.316 | 0.322 | 0.516 | 0.276 | −0.156 | −0.533 | 0.326 |
| Three treatments | −0.400 | 0.430 | 0.574 | 0.083 | −0.119 | −0.576 | 0.051 |
| Tebuconazole | −0.354 | 0.346 | 0.590 | 0.167 | −0.311 | −0.604 * | 0.302 |
| Epoxiconazole | −0.396 | 0.558 | 0.707 * | −0.051 | −0.172 | −0.412 | 0.135 |
| Epoxiconazole + Thiophanate-methyl | −0.444 | 0.503 | 0.550 | 0.045 | −0.041 | −0.527 | 0.181 |
| Strobilurin | −0.524 | 0.510 | 0.509 | −0.055 | −0.044 | −0.490 | 0.331 |

*—statistically significant coefficient at $p = 0.05$.

The dependence of root yield on total precipitations for individual months of the growing season was insignificant for the unprotected sugar beet (the control). The strongest (though also insignificant) negative correlation was found for June precipitation (r = −0.549) (Table 6). With fungicidal protection, however, this correlation was much weaker, especially after three treatments or one treatment with epoxiconazole, epoxiconazole + thiophanate-methyl, and strobilurin.

**Table 6.** Pearson's simple correlation coefficients between root yields and monthly sums of precipitation during the sugar beet growing season by fungicidal protection method.

| Protection Method | Month | | | | | | |
|---|---|---|---|---|---|---|---|
| | April | May | June | July | August | September | October |
| Control | 0.382 | 0.391 | −0.549 | −0.024 | 0.110 | −0.163 | 0.123 |
| Three treatments | 0.289 | 0.456 | −0.333 | 0.240 | 0.202 | 0.150 | 0.179 |
| Tebuconazole | 0.496 | 0.316 | −0.507 | 0.018 | 0.361 | 0.010 | 0.189 |
| Epoxiconazole | 0.306 | 0.282 | −0.316 | 0.187 | 0.213 | −0.040 | 0.330 |
| Epoxiconazole + Thiophanate-methyl | 0.283 | 0.356 | −0.328 | 0.110 | 0.080 | 0.041 | 0.150 |
| Strobilurin | 0.398 | 0.177 | −0.376 | −0.044 | 0.193 | 0.007 | 0.085 |

The multiple regression calculation showed that the yield of sugar beet roots depended strongly on weather patterns during the growing season. The yields of unprotected plants and those protected with the epoxiconazole + thiophanate methyl mixture was determined

to be 85.5% and 81.8%, respectively, by May and September's air temperatures and June's sum of precipitation (Table 7). For protected plants, the root yield was also determined by October temperature for the three treatments and for epoxiconazole and, even more so, by June temperature and April precipitation for tebuconazole. When strobilurin was applied to foliage, almost 100% of yield variability depended on air temperature in May–September (excluding June) and on precipitation in April, May, and June.

**Table 7.** Interdependent influence of air temperature and precipitation during the sugar beet growing season on root yield.

| Protection Method | Regression Equation | Correlation Coefficient | Determination Coefficient |
|---|---|---|---|
| Control | ˜ y1 = 6.587tV − 1.746tIX − 0.211oVI + 16.711 | 0.925 | 0.855 |
| Three treatments | y1 = 8.700tV − 2.503tIX − 3.463tX − 0.230oVI + 36.517 | 0.949 | 0.901 |
| Tebuconazole | y1 = 4.189tV + 1.874tVI − 2.207tIX − 1.782tX + 0.153oIV − 0.114oVI + 33.783 | 0.996 | 0.991 |
| Epoxiconazole | y1 = 8.863tV − 1.394tIX − 2.480tX − 0.216oVI + 5.787 | 0.952 | 0.906 |
| Epoxiconazole + thiophanate-methyl | y1 = 7.476tV − 1.927tIX − 0.160oVI + 8.210 | 0.905 | 0.818 |
| Strobilurin | y1 = 8770tV − 1.237tVII + 3.733tVIII − 1.702tIX + 0.316oIV + 0.057oV − 0.117oVI − 74.602 | 0.999 | 0.997 |

˜ variable designations—see Tables 1 and 2.

## 4. Discussion

The analysis of the climatic risk of plant cultivation in Kuyavia-Pomerania Voivodship indicates that the region where the field studies were conducted was characterized by highly variable meteorological conditions and weather patterns [30–32]. According to the cited authors, precipitation is the most variable in the warm half-year, which was also the case during the present study. However, even with periodic precipitation deficits, there was a progressive increase in sugar beet root yields, which is probably the result of biological and agrotechnical progress. A similar trend of >1.5 t·ha$^{-1}$ per year in sugar beet yield was found for north-eastern Poland in an earlier decade by Stępień [33]. Habitat conditions, especially precipitation and thermal conditions, determine not only root yield but also the need for and effectiveness of chemical plant protection, including those that combat fungal pathogens [34,35]. So too, the authors' research on sugar beet foliage [27] indicates that protecting leaves with fungicides reduces their infestation and damage by fungal pathogens, increases the LAI coefficient of the canopy (especially in the second half of the vegetation period), and increases leaf yield. However, the effectiveness of the fungicidal protection of leaves correlates strongly with rain and thermal conditions. Applying fungicides reduces the strength of the positive correlation between the amount of rainfall and the degree of beet leaf infection and damage.

Effective protection of beet's assimilative area against the effects of pathogens appears to be very important in the context of crop accumulation. This assumption is confirmed by the results of Moliszewska [36], which showed a significant correlation between root yield and the number of young leaves and then mature leaves in July and August. In integrated protection, after using non-chemical methods and treatments, foliar fungicides are also applied [37,38]. These preparations, properly selected and applied in accordance with the principles of good agricultural practice, improve the health of plants and reduce losses caused by diseases [39,40]. Their use in their own study area was justified. Firstly, in each year, the least affected and least damaged leaves were those of plants subjected to one or three fungicidal treatments, and the leaves of unprotected plants were the most affected. As a result of the effect that applying fungicide had on the condition of plant foliage, fungicidal protection resulted in a significant increase in leaf yield in ten out of eleven study years [27]. Secondly, this protection contributed to an increase of up to 21.8% in root yield on some plots. Such a relatively large increase in yield under the influence

of fungicidal treatments in sugar beet farming is confirmed in other national and global studies [41,42]. However, the occurrence and size of the yield protection effect depended on the type of active ingredients used, the number of such substances in a single treatment, and the number of treatments during the growing season. In six of the eleven study years, the root yield was significantly higher for three treatments than for at least one of the single treatments. However, only in one year did three applications of fungicides result in an increase in yield greater than all single treatments. Similar results were obtained by Kristek et al. [43] using one or three fungicide treatments against leaf spots in agricultural habitat conditions in Croatia. In the cited studies, the yield of roots after three treatments was higher than after a single application (though the extent differed between study years). In the cited studies, various fungicidal active substances, including systemic fungicides, were used to protect sugar beet. These included substances belonging to the same chemical groups as the preparations used in the present research, e.g., triazoles, strobilurins. Varying active ingredients between successive treatments or combining the use of several substances is justified by the desire to reduce the risk of facilitating the development of pathogenic resistance [44]. A long-term study of various active ingredients in many locations is also the basis for their comprehensive assessment and possible decisions to withdraw them from further use [45]. Of the active ingredients found in this study, thiophanate-methyl is the case.

The effect of fungicidal protection depends greatly on habitat conditions, especially weather patterns. In three out of eleven years, no significant differences in root yield were found as a result of the foliar application of fungicides. The effectiveness of fungicide application was found to be heavily dependent on climatic conditions also by Greiner et al. [46] and Juroszek et al. [47]. However, in our own research, this is confirmed by correlation calculations. This analysis showed, among other things, that when fungicidal protection is applied, beetroot yield has a lower negative correlation with June rainfall and a stronger positive correlation with air temperature in the same month than when no protection is applied.

## 5. Conclusions

The results of long-term field experiments conducted according to consistent methodological assumptions and in uniform soil conditions allow us to identify the need for the application of fungicidal protection to sugar beet leaves and how the yield-shaping efficiency of such treatments depends on local rainfall and thermal conditions. They, therefore, provide important information for formulating and verifying integrated production principles for this crop. The research allows us to conclude that, although foliar fungicidal protection has a positive effect on sugar beet root yields, in three out of the eleven years, the application of fungicides could have been foregone without significantly reducing the yield. In addition, in all but one year, selecting the appropriate active fungicidal ingredient or mixture of two such ingredients would have provided the same root yield as three fungicidal treatments would have. Decisions on the chemical protection of sugar beet leaves should involve ongoing monitoring and analysis of weather patterns during the vegetation period, especially in periods that are particularly important for the formation of the root yield. The experiments showed how fungicidal protection affected the dependence of root yield on variables characterizing the thermal and precipitation conditions during the beet growing season. Among other things, protected plants had a greater positive dependence of root yield on average June air temperature and a weaker negative correlation with total June rainfall than unprotected plants. The above conclusions, drawn on these long-term field experiments in a country that is one of the largest producers of sugar beet in Central and Eastern Europe and on the results of the cited studies of other authors, indicate the need to conduct permanent work to improve the principles of integrated sugar beet production, especially in terms of the rational reduction of pesticide use.



**Author Contributions:** Conceptualization, I.J., D.J. and J.K.; methodology, I.J., J.K. and D.J.; investigation, J.K., D.J. and I.J.; resources, J.K., M.R. and M.B.; data curation, J.K., I.J. and D.J.; formal analysis, I.J., J.K. and D.J.; writing—original draft preparation, I.J., D.J., J.K., M.R. and M.B.; writing—review and editing, I.J., D.J., J.K., M.R. and M.B. All authors have read and agreed to the published version of the manuscript.

**Funding:** This research received no external funding.

**Institutional Review Board Statement:** Not applicable.

**Data Availability Statement:** Not applicable.

**Acknowledgments:** The authors thank Nordzucker Polska S.A. for making field experiments possible.

**Conflicts of Interest:** The authors declare no conflict of interest.

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
