# Peer review of "Fungicidal Protection as Part of the Integrated Cultivation of Sugar Beet: An Assessment of the Influence on Root Yield in a Long-Term Study"

_agriculture, doi:10.3390/agriculture13071449_

Round 1
Reviewer 1 Report
In this manuscript authors reported the main results obtained in a long-term experiment to determine the effectiveness of fungicidal application in an integrated management strategy of sugar beet crop. This manuscript is with the scope of the journal Agriculture. However, before being published authors should correct or clarify several major issues.
Lines 21 to 24. Sentence should be revised. It is very hard to understand what means.
Abstract should include a conclusion of the study.
Line 25. Keywords should not be contented in title.
Lines 69 and 70. It is not clear what does ‘The experimental 69 factor was the method of fungicidal protection’ means? When authors say ‘Foliar fungicides were applied according to the experimental factor and its treatments’ in lines 101 and 102, I do not understand what the differences between the experimental factor and treatments are. How were products applied?
Lines 74 to75. I do not understand the different treatments. Did authors name ‘three treatments’ when they applied three different products one time each? Were the other products applied just one time?
Line 76. Did plot size include the four replicates? Which is the number of plants or size of each block or treatment?
Line 81. Delete ‘is’
Line 83. Indicate mean maximum and minimum temperature.
Line 105. When was the second treatment applied?
Line 107. Although pathogen infestation data had been previously published, they are one of the main support data of this study, so, at least, a brief summary should be included in this paper.
Line 109. I have a curiosity, because I did not know it: Which is the difference between the biological and technological content of sucrose?
In general authors should include more information about treatments application: doses, form of application, number of plants treated per each treatment, and so on.
It is not clear if authors evaluated any variable in this study or they are going to do it in future (lines 109 to 112).
Line 114. Which results were analyzed? Authors did not specify.
How did authors determine the root yield? Which methodology did they use to qualify root? This information is missing.
Where and how did authors obtain thermal and precipitation conditions of the study area?
Statistical analysis section is very difficult to understand because the rest of the material and methods section is not clear. Authors should deeply improve this section to facilitate readers’ comprehension.
In general, material and methods section required a deeply reorganization.
Table 1 and 2. What is the difference between mean and long-term mean temperature?
To understand the results of root yield after treatments application is crucial to know the data about pathogen incidence in the study plots every year, because disease incidence is also depended on climatic conditions.
Table 3. Include in the footnote more information about the statistical analysis and the test used for the mean values comparison.
In table 3, did authors do the mean values comparison among treatments for each year? When there are not significant differences among treatments, letters are not necessary (for example, year 2006, delete ‘a’ in all treatments).
Lines 228 to 230. We do not have any data about pathogen damages in plants in this study.
Line 231. Authors did not show results about leaf yield, only root.
Information in line 242 is repetitive.
Is June a critical month for sugar beet diseases development? Because authors focused their conclusion only on weather parameters, but they did not relate with disease incidence.
Some conclusions reported in discussion section are not completely based on results exposed in this study.
Author Response
The authors are grateful for review of the manuscript. The article was supplemented and corrected in accordance with the comments and suggestions of the reviewer. Among others: - a correction to the summary was made, - this part of the article was supplemented with a general conclusion of the research, - keywords have been corrected, The Materials and Methods section has been supplemented and corrected: - the procedures of the experiment were described in detail, - supplemented meteorological data in the study area, - added information about the source of meteorological data, - supplemented information on the application of fungicides, - it was indicated that more information about the conditions and principles of conducting a field experiment can be found in an earlier scientific article, - some results and conclusions from the previous article were recalled (occurrence of pathogens, infection and damage to sugar beet leaves), - added information on how to harvest beet roots and determine the final yield, - the method of statistical analysis of the results was presented more precisely. In the following chapters of the article, the authors: - indicated the difference between mean and long-term mean temperature and corrected table 1 and table 2 accordingly, - corrected table 3 - added information about the statistical test to table 3 and figure 1, - introduced corrections and additions to the Discussion chapter. In the prepared version of the article, the linguistic and formal errors indicated by the reviewers were also taken into account. Corrections in the manuscript are in red font.

Reviewer 2 Report
row 42: mankozeb instead mencozeb
it would be interesting to compare not just the root yield, but also the rot sucrose content
Author Response
The authors are grateful for two reviews of the manuscript. The article was supplemented and corrected in accordance with the comments and suggestions of the both reviewers. Among others: - a correction to the summary was made, - this part of the article was supplemented with a general conclusion of the research, - keywords have been corrected, The Materials and Methods section has been supplemented and corrected: - the procedures of the experiment were described in detail, - supplemented meteorological data in the study area, - added information about the source of meteorological data, - supplemented information on the application of fungicides, - it was indicated that more information about the conditions and principles of conducting a field experiment can be found in an earlier scientific article, - some results and conclusions from the previous article were recalled (occurrence of pathogens, infection and damage to sugar beet leaves), - added information on how to harvest beet roots and determine the final yield, - the method of statistical analysis of the results was presented more precisely. In the following chapters of the article, the authors: - indicated the difference between mean and long-term mean temperature and corrected table 1 and table 2 accordingly, - corrected table 3 - added information about the statistical test to table 3 and figure 1, - introduced corrections and additions to the Discussion chapter. In the prepared version of the article, the linguistic and formal errors indicated by the reviewers were also taken into account. Corrections in the manuscript are in red font.

Round 2
Reviewer 1 Report
Authors took in consideration the suggestions and manuscript is ready to be published.